# Visualizing a field of research: A methodology of systematic scientometric reviews

Chaomei Chen[1,2]*, Min Song[2]

**1** Department of Information Science, College of Computing and Informatics, Drexel University, Philadelphia, Pennsylvania, United States of America, **2** Department of Information Science, Yonsei University, Seoul, Republic of Korea

* chaomei.chen@drexel.edu

**Data Availability Statement:** All relevant data are within the manuscript, Supporting Information files, and on Figshare: https://doi.org/10.6084/m9.figshare.9939773.v1.

**Funding:** CC acknowledges the support of the SciSIP Program of the National Science Foundation

## Abstract

Systematic scientometric reviews, empowered by computational and visual analytic approaches, offer opportunities to improve the timeliness, accessibility, and reproducibility of studies of the literature of a field of research. On the other hand, effectively and adequately identifying the most representative body of scholarly publications as the basis of subsequent analyses remains a common bottleneck in the current practice. What can we do to reduce the risk of missing something potentially significant? How can we compare different search strategies in terms of the relevance and specificity of topical areas covered? In this study, we introduce a flexible and generic methodology based on a significant extension of the general conceptual framework of citation indexing for delineating the literature of a research field. The method, through cascading citation expansion, provides a practical connection between studies of science from local and global perspectives. We demonstrate an application of the methodology to the research of literature-based discovery (LBD) and compare five datasets constructed based on three use scenarios and corresponding retrieval strategies, namely a query-based lexical search (one dataset), forward expansions starting from a groundbreaking article of LBD (two datasets), and backward expansions starting from a recently published review article by a prominent expert in LBD (two datasets). We particularly discuss the relevance of areas captured by expansion processes with reference to the query-based scientometric visualization. The method used in this study for comparing bibliometric datasets is applicable to comparative studies of search strategies.

## Introduction

Systematic reviews play a critical role in scholarly communication [1]. Systematic reviews typically synthesize findings from original research in a field of study, assess the degree of consensus or the lack of it concerning the state of the art in the field, and identify challenges and future directions. For newcomers to a field of study, a timely and comprehensive systematic review can provide a valuable overview of the intellectual landscape and guide new researchers to pursue their research effectively. For experienced and active researchers, systematic reviews can be instrumental in keeping their knowledge of the field up to date, especially when

(Award #1633286), the support of Microsoft Azure Sponsorship. Data sourced from Dimensions, an inter-linked research information system provided by Digital Science (https://www.dimensions.ai). MS acknowledges the support of the Ministry of Education of the Republic of Korea and the National Research Foundation of Korea (NRF-2018S1A3A2075114) and partial support from the Yonsei University Research Fund of 2019-22-0066. The funders had no role in study design, data collection and analysis, decision to publish, or preparation of the manuscript.

**Competing interests:** The authors have declared that no competing interests exist.

involving areas that are potentially relevant but fall outside the immediate topic of one's interest. Researchers have also studied scientific literature to uncover potentially significant but currently overlooked connections between disparate bodies of literature, notably, as demonstrated in the research of literature-based discovery (LBD) [2–7].

A fast-growing trend is the increase of systematic reviews conducted with the assistance of science mapping tools [8]. A science mapping tool typically takes a set of bibliographic records of a research field and generates an overview of the underlying knowledge domain, e.g. as with CiteSpace [9, 10] and VOSviewer [11]. For example, systematic reviews facilitated by using CiteSpace include a diverse range of research areas such as regenerative medicine [12], natural disaster research [13], greenhouse gas emission [14], and identifying disruptive innovation and emerging technology [15]. Similarly, VOSviewer was used in reviews of topics such as citizen science [16] and climate change [17]. A scientometric overview of a field of research provides a valuable source of input to conducting systematic reviews, especially in situations when relevant and up-to-date systematic reviews may not be readily available or accessible. The quality of the input data therefore is critical to the overall quality of subsequent analyses and reviews. This practical issue is particularly significant when we need to select subsets of articles from a large pool of available data or when we want to limit the scope of a study to specific disciplines as opposed to open to all disciplines.

Considerations concerning the scope of selection have been discussed in the literature in terms of local, global, and hybrid approaches [18]. Global maps of science by definition provide a comprehensive coverage of all scientific disciplines [19], whereas local maps typically focus on selected areas of interest. Hybrid maps may use a global map as a base map and superimpose local maps as overlays [20]. Generating global maps of science requires a substantial array of resources that are not commonly accessible to most of researchers, whereas resources required for generating local maps are more reachable, especially with the increasing accessible science mapping tools. Global maps are valuable as they provide a broad context of a specific research interest. In practice, generating global maps is resource consuming and demanding, for example, requiring direct access to large data sources that are only accessible to a small number of researchers. Consequently, global maps may not be updated as frequently as needed by end users. For example, a significant update of a global model may not be undertaken for 5 years [20]. The underlying structure of a research field is often subject to a variety of changes as the scientific literature grows over time [21, 22]. Therefore, it is reasonable to question whether an existing global model created a few years ago remains a valid representation of the underlying structure for specific analytic tasks in hand, although the answer may differ at different levels of granularity.

In contrast, approaches of localism face different challenges. For example, a common challenge for individual researchers of science mapping tools such as CiteSpace [9, 10], VOSviewer [11], and HistCite [23] is to select a representative subset of articles from tens of millions or more of scholarly publications in the Web of Science, Scopus, Dimensions, and/or the Lens. Researchers have used lexical search, citation expansion, and hybrid strategies [24–26].

Complex and sophisticated queries are typically constructed with the input from domain experts and iteratively refined as demonstrated in several in-depths studies [25–27]. We refer such strategies as query-based approaches in this article. The role of relevant and sufficient domain expertise in such approaches is critical, which may lead to a double-edged sword. As we will see in this article, using computational approaches to reduce the cognitive burden from domain experts is a noticeable area of research. Following the principles of LBD to emphasize potentially valuable but currently overlooked connections in scientific literature, it is conceivable that there are undiscovered connections where even domain experts' input may

be limited. Reducing the initial reliance on domain expertise can increase the applicability of query-based strategies.

In this article, we propose a flexible computational approach to the construction of a representative dataset of scholarly publications concerning a field of research through iterative citation expansions. Starting from an initial dataset or even a single seed article, the expansion process will automatically expand the initial set by adding articles through citation links in forward, backward, or both directions. This flexibility is particularly valuable in some common scenarios in practice. For example, Swanson's article on fish oil and Raynaud's syndrome [6] is widely considered as a groundbreaking one in LBD research. A researcher may want to retrieve subsequently published articles that are connected to the groundbreaking article through potentially lengthy citation paths. Such expansions are valuable to preserve the continuity of research literature identified across an extensive time span. In contrast, a researcher may come across a recently published review article of LBD and would like to find previously published articles that lead to the state of the knowledge summarized in the review. For example, in 2017, Swanson's long-term collaborator Smalheiser published an article entitled "Rediscovering Don Swanson: the past, present and future of literature-based discovery" [28] and researchers may be interested in tracing several generations of relevant articles.

Pragmatically speaking, the ability to expand any set of articles of interest provides a smooth transition across a local-global continuum. The agility of the approach enables us to improve the quality of an input dataset of scientometric studies by refining the context incrementally. We demonstrate the expansion and evaluation of five datasets retrieved by different strategies on literature-based discovery (LBD). The field of LBD is potentially very broad as it is applicable to numerous scientific disciplines. Swanson's pioneering studies in the 1980s have been a major source of inspiration [6, 7, 29, 30], followed by a series of studies in collaboration with Smalheiser [31–33], who recently reviewed the past, present, and future of LBD [28]. Significant developments have also been made by other researchers along this generic framework, for example, [34–36]. It would be challenging to formulate a complex query to capture a diverse range of relevant research activities, some of which may have drifted away from the core literature of LBD.

Judging the relevance of topics is often situational in nature and therefore challenging even with domain expertise. In LBD, the threshold of relevance is by definition lower than other fields because the focus is on undiscovered connections, which in turn leads to a demand for an even higher level of recall for conducting systematic scientometric reviews. Shneider proposed a four-stage model to characterize how a scientific field may evolve, namely, identifying the problem, building tools and instruments, applying tools to the problem, and codifying lessons learned [22, 37]. The application stage may also reveal unanticipated problems, which in turn may lead to a new line of research and form a new field of research. The LBD research may continue along the research directions set off by the pioneering studies in LBD. According to Shneider's four-stage model, new specialties of research may emerge. A systematic review of LBD should open to such new developments as well as the established ones that we are familiar with. What can we gain by using cascading citation expansions? What might we miss if we rely on the simple query-based search strategy alone? What types of biases may be introduced or avoided by cascading expansions as well as by selecting points of departure?

To address these questions, we develop an intuitive visual analytic method to compare multiple search strategies and reveal the strengths and weaknesses of a specific procedure. We compare a total of five datasets in this study, including a dataset from a simple query with phrases of "literature-based discovery" as a baseline reference and four datasets from cascading citation expansions based on two singleton seed article sets using Dimensions as the source of data.

The rest of the article is organized as follows. We characterize existing science mapping approaches in terms globalism and localism, especially their strengths and weaknesses for conducting systematic reviews of relevant literature. We introduce a flexible citation expansion approach–cascading citation expansion–to improve the data quality for conducting systematic scientometric reviews. We first demonstrate what a query-based strategy may reveal about the LBD research, then we combine the five datasets to form a common baseline for comparing the topics covered by the five individual datasets.

## Mapping the scientific landscape

Considerations concerning the scope of a study of scientific literature can be characterized as global, local, and hybrid in terms of their intended scope and applications.

### Globalism

Global maps of science aim to represent all scientific disciplines [18]. Commonly used units of analysis in global maps of science include journals and, to a much less extent, articles. Clusters of journals are typically used to represent disciplines of science.

Global maps of science are valuable for developing an understanding of the structure of scientific knowledge involving all disciplines. Global maps provide a relatively stable representation of the underlying structure of knowledge, which can be rather volatile at finer levels of granularity. Klavans and Boyack compared the structure of 20 global maps of science [38]. They arrive at a consensus map generated from edges that occur in at least half of the input maps. A stable global representation is suitable to serve the role of an organizing framework to accommodate additional details as overlays [19, 20, 39]. Global maps may reveal insights that may not be possible at a smaller scale. For example, in studies of structural variations caused by scholarly publications, detecting a potentially transformative link in a network representation of the literature relies on the extent to which the contexts of both ends of the link are adequately represented [21].

On the other hand, the validity of a global map as a base map is likely to decrease over time as subsequent publications may change the underlying knowledge structure considerably [21]. Given that updating a global map of science is a very resource demanding task and it may be years before a major update becomes available [20], is the structure shown in a global map still a valid representation at the intended level of granularity? If a global map is updated too often, it may lose its stability value as an organizing framework to sustain overlays. A fundamental question is not necessarily whether the representation is categorically comprehensive, but whether the structural representation as a context for analyzing the literature is adequate and effective.

### Localism

To localism, the focus is on communicating the structure of a subject matter of interest at the level of scientific inquiry and scholarly communication, including analytic reasoning, hypothesis generation, and argumentation. Many studies of science and systematic scientometric reviews belong to this category. Scientometric studies commonly draw bibliographic data, especially citation data, from long-established sources such as the Web of Science and Scopus. More recent additions include Microsoft Academic Search, Dimensions, and the Lens. Knowledge domain visualization, for example, focuses on knowledge domains as the unit of analysis [10, 40, 41]. The concept of a knowledge domain emphasizes that the boundary of a research field should reflect the underlying structure of knowledge. Relying on organizing structures

such as institutions, journals, and even articles along has the risk of missing potentially significant articles.

Query-based search is perhaps by far the most popular strategy for finding articles relevant to a topic of interest or a field of research. A query-based search process typically starts with a list of keywords or phrases provided by an end user. For example, a query of "*literature-based discovery*" OR "*undiscovered public knowledge*" could be a valid starting point to search for articles in the field of LBD. The quality of retrieval is routinely measured in terms of recall and precision. Initial search for systematic reviews tends to give a higher priority to recall than precision.

Formulating a good query is a non-trivial task even for a domain expert because the quality of a query can be affected by several factors, including our current domain knowledge and our motivations of the search [24]. Furthermore, if our goal is to identify emerging topics and trends in a research field, which is very likely when we conduct a systematic review of the field, then it would be a significant challenge to formulate a complex query effectively in advance. As a result, iteratively refining queries over lessons learned from the performance of previous queries is a common strategy, especially when combined with the input from relevant domain experts [25, 26]. Scatter/Gatherer [42] is an influential early example of iteratively improving the quality of retrieved information based on feedback.

A profound challenge to query-based search is the detection of an implicit semantic connection, or a latent semantic relation. Search systems have utilized additional resources such as WordNet [43] and domain ontologies [44, 45] to enhance users' original terms with their synonyms and/or closely related concepts. With techniques such as latent semantic indexing [46] and more recent advances in distributional semantic learning [47], the relevance of an article to a topic can be established in terms of its distributional properties of language use. The semantic similarity of an article can be detected without an explicit presence of any keywords from users' original query. This is known as the distributional hypothesis: linguistic items with similar distributions have similar meanings.

## Cascading citation expansion

In this article, we conceptualize a unifying framework that accommodates both globalism and localism as special cases on a consistent and continuous spectrum through a combination of query-based search and cascading citation expansion. Under the new conceptualization, the coverage of a local map can grow incrementally as the expansion process may continue as many generations of citation as needed. Thus, the gap between a local map and a global map can be reduced considerably. Table 1 summarizes the major advantages and weaknesses of globalism and localism along with potential benefits that the conceptualized incremental expansion may provide.

In this article, we demonstrate the flexibility and extensibility of a incremental expansion approach–cascading citation expansion–by applying this methodology to a few common scenarios in research in relation to a field of research of own interest, i.e. literature-based discovery (LBD).

Citation indexing was originally proposed by Eugene Garfield to tackle the information retrieval problem in the context of scientific literature [48]. While the nature of a citation may vary widely as many researchers have documented [49], an instance of a citation from one article to another provides evidence of some potentially significant connections. A unique advantage of a citation-based search method is that it frees us from having to specify a potentially relevant topic in our initial query, which is useful to reduce the risk of missing important relevant topics that we may not be aware of.

**Table 1. Contrasts between globalism and localism approaches.**

| Criteria for representing a research field | Globalism | Localism | Incremental Expansion |
|---|---|---|---|
| Broad context | High | Low | Increased |
| Coverage, Diversity, Recall | High | Low | Increased |
| Redundancy | High | Low | Reduced |
| Structural stability | High | Low | Increased |
| Precision | Low | High | Increased |
| Timeliness | Low | High | Increased |

We often encounter situations in which we have found a small set of highly relevant articles and yet they are still not fully satisfactory for some reasons. For example, if we were new to a topic of interest and all we have to start with is a systematic review of the topic published many years ago, how can we bring our knowledge up to date? If what we have is a newly published review written by a domain expert, how can we expand the review to find more relevant articles? Another common scenario is when we want to construct a relatively comprehensive survey of a target topic, how can we generate an optimal dataset that is comprehensive enough but also contains the least amount of less relevant articles.

When we encounter these situations, an effective method would enable us to build on what we have found so far and add new articles iteratively. Scatter/Gather [42] was a dynamic clustering strategy proposed in mid-1990s by allowing users to re-focus on query-specific relevant documents as opposed to query-independent clusters in browsing search results. From the citation indexing point of view, articles that cite any articles in the initial result set are good candidates for further consideration. Thus, an incremental expansion strategy can be built based on these insights to uncover additional relevant articles.

Incremental expansion processes can be performed in parallel or in sequence. When an expansion step is applied to a base set S of articles, articles associated with S through citation links are added to S thus expand the set S. Users may define an inclusion threshold such that the expansion is limited to adding articles with sufficient citations. We refer successive citation expansions as cascading citation expansion [50]. The concept of cascading citation expansion is intrinsic to the general framework of citation indexing. An expansion process is capable of generating an adequate context for a systematic scientometric review of a research field by incrementally retrieving more and more articles from the literature.

Cascading citation expansion requires a constant programmatic access to a master source of scientific articles. We utilize the Dimensions API to access their collection of over 98 million publications (at the time of writing). A cascading citation expansion process may move forward and backward along citation paths. If the entire universe of all the scientific publications is completely reachable from one article to another, then cascading citation expansion will eventually reach all the publications, which means that one can achieve a coverage as large as we wish. If the entire universe is in fact made of several galaxies that are not reachable through citation links, the question is whether this is desirable to transcend the void and reach other galaxies. Strategies using hybrid lexical and semantic resources may become useful in such situations, but in this study we assume it is acceptable to terminate the expansion process once citation links associated with a chosen starting point are exhausted.

Fig 1 illustrates the cascading citation expansion process. Given an initial set of seed articles, which can be a singleton set, a 1-generation forward citation expansion will add articles that cite any member article of the seed set directly, i.e. with a one-step citation path. A 2-generation forward expansion will add articles connecting to the seed set with two-step citation paths. We include a 5-generation forward expansion in this study.

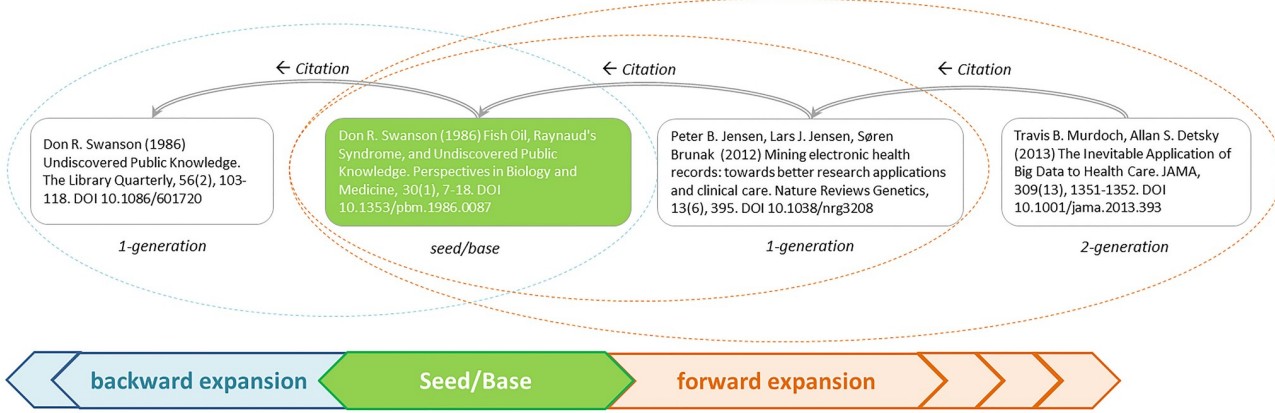

**Fig 1. Applying incremental citation expansions increases the quality of input data for mapping a research field.**

## Use scenarios and corresponding search strategies

Here we consider two common scenarios in research. In the first scenario, we have identified a well-known classic work, for example, Swanson's 1986 article on fish oil and Raynaud's syndrome [6], and would like to retrieve all follow-up studies and articles that cited the original work directly or indirectly since 1986. What are the newly developed major topics ever since? What are the hottest and the most far-reaching topics in more recent years? Are there any areas branching off the main paths?

In the second scenario, we are attracted to a very interesting recently published article and we would like to collect relevant studies in the past that lead to the article, i.e., its intellectual base. Smalheiser's 2017 review of LBD [28] is an example. Smalheiser has co-authored with Swanson on several landmark studies in the development of literature-based discovery. Smalheiser cited 71 references in his review. What would be a broader context of Smalheiser's 71-reference review? Are there LBD-relevant topics but excluded by the authoritative domain expert?

Pragmatically, if we were to rely on the simple full text search alone, how much would we miss? Are there topics that we might have missed completely? What would be an optimal search strategy that not only adequately captures the essence of the development of the field but also does in the most efficient way? Borrowing the terminology from information retrieval, an optimal search strategy should maximize the recall and the precision at the same time.

Cascading citation expansion functions are implemented in CiteSpace based on the Dimensions' API. The expansion process starts with an initial search query in DSL, which is Dimensions' search language. Users who are familiar with SQL should be able to recognize the resemblance immediately. The result of the initial query forms the initial set of articles. In fact, in addition to publications, one can retrieve grants, patents, and clinical trials from Dimensions. In this study, we concentrate on publications.

## Constructing five datasets of literature-based discovery

To demonstrate the flexibility and extensibility of the incremental expansion approach, we take the literature-based discovery (LBD) research as the field to study. We choose LBD for several reasons: 1) we are familiar with the early development of the domain, 2) we are aware of a recent review written by one of the pioneer researchers and we would like to set it in a broader context, and 3) we would like to take this opportunity to demonstrate how one can

**Table 2. Five datasets on literature-based discovery.**

| Set | Description | Records |
|---|---|---|
| F | Fulltext search on Dimensions for "*literature-based discovery*" OR "*undiscovered public knowledge*" (Data of Search: 3/4/2019) | 1,777 |
| S₃ | 3-generation forward expansion from a seed article by Swanson [6] Dimensions:pub.1053548428 (Times Cited: 407) Citing Article and Reference Thresholds: 10, 10 | 748 |
| S₅ | 5-generation forward expansion of Swanson's pioneering work [6] Citing Article and Reference Thresholds: 20, 20 | 45,178 |
| N_F | Forward expansion from a 2017 seed article by Smalheiser [28] | 73 |
| N_B | Backward expansion from N_F | 2,451 |
| | Combined | 48,298 |

apply the methodology to a visual exploration of the relevant literature and develop a good understanding of the state of the art. These reasons echo the common scenarios discussed earlier.

Table 2 summarized the construction of the five datasets included in the study, including key parameters such as citation thresholds.

Fig 2 illustrates the process of the comparative study of five datasets retrieved based on a query-based search and cascading citation expansions. In this study, we applied a citation filter in cascading citation expansions to the selection of citing and cited articles. Articles with citations below the threshold are filtered out from the expansion processes. These filters provide users with a flexible trade-off option between concentrating on major citation paths with a reduced completion time versus retrieving articles comprehensively with a much

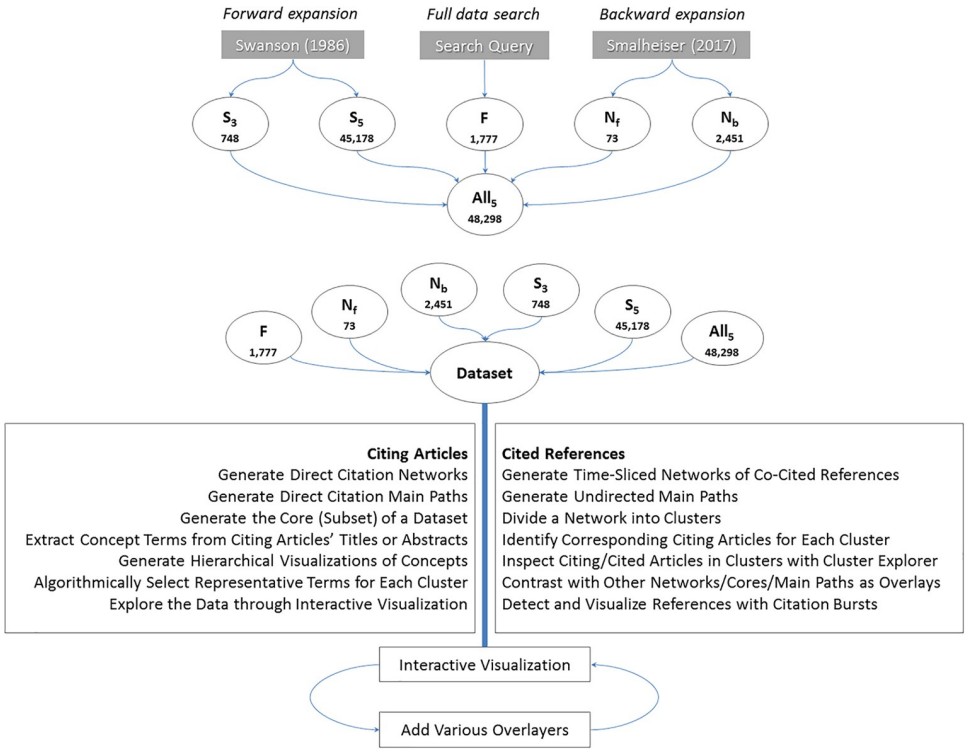

**Fig 2. Using multiple search and expansion strategies improves the data quality for scientometric studies.**

longer completion time. Since the distribution of citations of articles follows power law, a comprehensive expansion process may become too long to be viable for a daily use of these functions. The DSL query searched for Swanson's two articles published in 1986, namely, the fish oil and Raynaud' syndrome article 1986a [6] and the undiscovered public knowledge article 1986b [7]. The query search found 1,777 articles as the set F for Full data search. Swanson 1986a is used as the seed article for two multi-generation forward citation expansions, one for 3 generations (as set $S_3$) and the other for 5 generations (as set $S_5$). $S_3$ contains 748 articles, whereas $S_5$ is about 60 times larger, containing 45,178 articles. The other two datasets are expanded from Smalheiser's 2017 review as the seed, $N_F$ and $N_B$, where N is for Neil, Smalheiser's first name. Smalheiser's review contains 71 references. At the time of the experiment, a forward expansion from it found two articles that cite the review. The 73 articles form the set $N_F$. The $N_B$ set is obtained by applying backward citation expansions on the set $N_F$. The expansion stopped in 1934 with 2,451 articles. The five datasets are combined as the set $All_5$, containing 48,298 unique articles. $S_5$ contributed most of articles to the combination. The five datasets overlap to a different extent. S3 is a subset of S5. NB expands from NF. F overlaps with S5 the most (702 articles out of its 1,777 articles). Each of the five individual datasets and the combined dataset are visualized in CiteSpace as networks of co-cited references with thematic labels for clusters.

Given a dataset, its core is defined by references that satisfy two conditions, inspired by [24]: 1) global citation scores (GCS) in Dimensions are greater than two and 2) the ratio between local citation scores (LCS) within the analyzed dataset to corresponding GCS are greater than or equal to 0.01. According to [24], the core represents articles with a sufficient specificity to the field of research in question. The core will downplay the role of an article that has a very high GCS but a low LCS because it suggests that the article probably belongs to a field elsewhere. It may be possible for an article to have its citations evenly split across multiple fields, but its LCS/GCS ratio in any of these fields should have a good chance to qualify the article for the core.

Researchers have used main paths of a citation network to study major flows of information or the diffusion of ideas [51, 52]. Main paths of a dataset are derived from the corresponding direct citation networks of the dataset. Direct citation networks are generated in CiteSpace with GCS of 1 as the selection threshold. Pajek is used to select main paths based on Search Path Link Count (SPLC) using top 30 key routes found by local search.

Datasets, their cores, main paths, and clusters can be used as the initial seed set for cascading citation expansion. They can all be used as network overlays in CiteSpace to delineate the scope of a research field at various levels of granularity.

Fig 3 shows logarithmically transformed distributions of the five datasets. The distributions shown under the title are the original ones.

- The F dataset (in blue) is evenly distributed except a surprising peak in 2009, which turns out due to many articles from an encyclopedia. The number of articles each year ranges between 60 and 130.

- Both $S_5$ (red) and $S_3$ (orange) are forward expansions starting with Swanson's 1986 article on fish oil and Raynaud's syndrome [6]. In $S_3$, the inclusion threshold was at least 10 citations, whereas it was 20 in $S_5$ so as to keep the total processing time down. The majority of the articles in $S_3$ appeared between 2006 and 2018. $S_3$ had the first peak in 1984. It didn't return to the same level for the next 10 years until it started to climb up from 2003 and reached the second peak in 2012. In contrast, the distribution of the more extensive forward expansion $S_5$ shows a steady increase all the way over time.

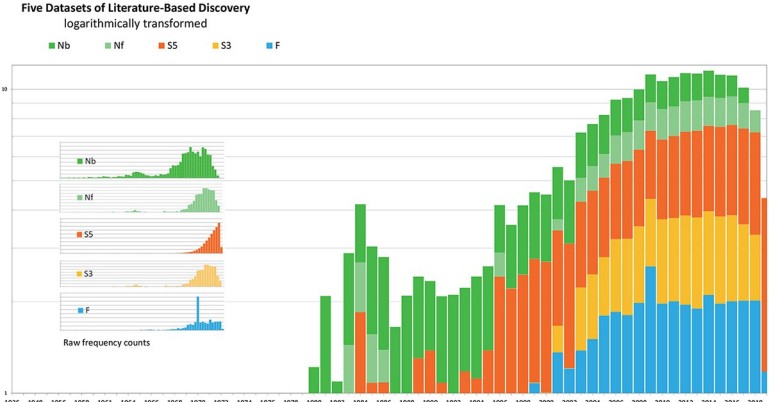

**Fig 3. Logarithmically transformed distributions of articles by year in the five datasets.**

- $N_F$, a forward expansion from Smalheiser's review of Swanson's work [28], includes articles that cited the references used in Smalheiser's review. Its distribution steadily increased between 1986 and 2017 with a rise of 5 over the 31-year span till 2016.

- $N_B$, a backward expansion from the set $N_F$, contains articles that are cited by $N_F$. The earliest article in $N_B$ was published in 1936. A noticeable hump from 1983 followed by a valley around early 1990s. Two peaks appeared in 2006 and 2011, respectively.

We will focus on how these datasets differ in terms of networks of co-cited references in the following analysis. There are of course many other ways to conduct scientometric studies based on these datasets, but we will limit to the co-citation networks generated with CiteSpace. It is important to note that co-citation networks in CiteSpace include much more information than a classic co-citation network, notably including various indicators and thematic labels derived from citing articles to clusters of co-cited references, year-by-year concept labels to track the evolution of a cluster, and a hierarchical representation of concept terms extracted from citing articles' titles and abstracts. Some of these features will be illustrated in the following sections.

The five datasets and the combined dataset are processed in CiteSpace with the consistent configurations. In particular, the link-to-node ratio is 3, look back years is 10, annual citation threshold is 2, and the node selection is based on g-index with a scaling factor of 30. These configuration settings are derived empirically, which tend to identify meaningful patterns. Given a dataset, CiteSpace first develops a network model by synthesizing a time series of annual networks of co-cited references. Then the synthesized network is divided into clusters of cited references. Themes in each clusters are identified based on noun phrases extracted from citing articles' titles and abstracts. Citing articles to a cluster are defined as articles that cite at least one member of the cluster. Extracted noun phrases are further computed to identify the most representative ones as the thematic labels for their cluster. CiteSpace supports three ways to select cluster labels based on Latent Semantic Indexing, Log-Likelihood Ratio Test, and Mutual Information [53]. CiteSpace supports a built-in database. Many attributes of datasets can be compared with the database. The core of a dataset, for example, can be identified using SQL queries.

Interactive visualizations in CiteSpace support several views, i.e., types of visualization, including a cluster view, a timeline view, a history view, and a hierarchical view. A network can be superimposed to another network as a layer. A list of references can be superimposed to a network as well. We use this feature to overlay the core and main paths of a dataset to its own

**Table 3. Properties of datasets and their networks (1986–2019).**

|  | *Combined* | *F* | *$N_F$* | *$N_B$* | *$S_3$* | *$S_5$* |
|---|---|---|---|---|---|---|
| **Citing Articles** | 48,298 | 1,777 | 73 | 2,451 | 748 | 45,178 |
| **References** | 1,162,614 | 47,416 | 2,862 | 70,489 | 6,601 | 1,071,079 |
| **Unique References** | 507,349 | 30,606 | 2,239 | 47,252 | 3,933 | 454,732 |
| **Core References** | 15,254 | 367 | 59 | 1685 | 86 | 13,231 |
| **Nodes** | 3,095 | 1,269 | 571 | 1,644 | 649 | 2833 |
| **Links** | 16,314 | 5,937 | 1,860 | 6,421 | 2,123 | 15,368 |
| **LCC (%)** | 2,397 (77) | 1,029 (74) | 257 (45) | 1,008 (61) | 461 (71) | 2,386 |
| **Modularity** | 0.84 | 0.76 | 0.93 | 0.87 | 0.71 | 0.83 |
| **Silhouette** | 0.34 | 0.34 | 0.50 | 0.44 | 0.47 | 0.29 |

CiteSpace Configuration: Network: LRF = 3, LBY = 10, e = 2.0, g(30); Core: GCS>2; LCS/GCS≥ 0.01.

network or to a network of another dataset. CiteSpace reports network and cluster properties such as modularity and silhouette scores. The modularity score of a network reflects the clarity of the network structure at the level of decomposed clusters. The silhouette score of a cluster measures the homogeneity of its members. A network with a high modularity and a high average of silhouette scores would be desirable. We will focus on the largest connect component of each network, which is shown as the default visualization. Users may choose to reveal all components of a network if they wish.

Table 3 summarizes various properties of the five individual datasets and the combined set along with their networks and core references. The F dataset, for example, contains 1,777 articles, which in turn cite 30,606 unique references. Among them, 367 references are identified as the core references based on the LCS/GCS ratio and the threshold of 3 for GCS. The resultant network contains 1,269 references as nodes and 5,937 co-citation links. The largest connected component (LCC) consists of 1,029 references, or 74% of the entire network. The modularity with reference to the clusters is 0.76, indicating a relatively high level of clarity. The average silhouette score of 0.34 out of 1.0 is moderate.

The modularity of a network measures the clarity of the network structure in terms of how well the entire network can be naturally divided into clusters such that nodes within the same cluster are tightly coupled, whereas nodes in different clusters are loosely coupled. The higher the modularity is, the easier to find such a division. The silhouette score of a network measures the average homogeneity of derived clusters [53]. The higher the average silhouette score is, the more meaningful a group is in terms of a cluster. The smallest dataset $N_F$ has the highest modularity of 0.93. It also has the highest average silhouette score of 0.50. The full text search has the modularity of 0.76, which is slightly lower than $N_B$, but its silhouette value of 0.34 is lower than others except $S_5$.

## Literature-based discovery

In this section, we visualize the thematic landscape of the field of literature-based discovery from multiple perspectives of the five datasets. We will start with the full text search results and then cascading citation expansions.

### Full text search

The query for the full text search on Dimensions consists of 'literature-based discovery' and 'undiscovered public knowledge.' The phrase 'literature-based discovery' is commonly used as the name of the research field. The phrase 'undiscovered public knowledge' appears in the

titles of two Swanson's publications in 1986. One is entitled "Fish oil, Raynaud's Syndrome [6], and Undiscovered Public Knowledge" in Perspectives in Biology and Medicine and the other is "Undiscovered Public Knowledge" in Library Quarterly [7].

The full text search found 1,777 records. Dimensions' export center supports the export of up to 50,000 records to a file in a CSV format for CiteSpace [9, 40]. Publication records returned from Dimensions do not include abstracts. We found 431 matched records in PubMed with their abstracts, but in this study the analysis is based on the full set of 1,777 records regardless they have abstracts or not because we primarily focus on the references they cite.

Fig 4 shows an overview map of LBD according to the dataset F. The color of a link indicates the earliest year when two publications were co-cited for the first time in the dataset. In this visualization, the earliest work appeared from the top of the network, whereas the most recent ones appeared at the bottom, although CiteSpace does not utilize any particular layout mechanisms to orient the visualization. The network is decomposed into clusters of references based the strengths of co-citation links. Clusters are numbered in the descending order of their size. The largest one is numbered as #0, followed by #1, and so on. Fig 4 also depicts the core of the F set as an overlay (in green) and its main paths (in red).

The largest cluster is #0 machine learning. More recent clusters, further down in the visualized network, include #1 semantic predication, and #7 citation network.

Fig 5 shows more features of the dataset F through colormaps of time (a), an overlay of its core (b), its and main paths (c). Labeled nodes on main paths include Swanson DR (1986), which also appears to be one of the oldest core references, Swanson DR (1997), Smalheiser (1998), Webber W (2001), and the most recently Cohen T (2010).

Fig 5 also contrasts references cited by Smalheiser's 2017 review of LBD and the main paths (d) and references cited by another LBD review published in 2017 by Sebastian et al. [5] in part (e). There are several notable differences. Torvik VI (2007) cited in Smalheiser's review was not on the main paths. In contrast, a few articles on the main paths such as Hunter L (2006), Zweigenbaum P (2007), and Agarwal P (2008) are not cited in Smalheiser's review. Similarly, Sebastian et al.'s review also cited Torvik VI (2007) and a few other articles off the main paths, including Kostoff RN (2009), Chen C (2009), and Kostoff RN (2008). The general area of Sebastian et al.'s review overlaps with that of Smalheiser's review considerably except Sebastian et al.'s review reached further towards cluster #7 citation network. The boundaries of clusters are show in distinct colors in part (f) of Fig 5. According to the colored cluster areas, the main paths go through #0 machine learning, whereas the two LBD reviews did not.

In Fig 6, references cited by the two LBD review articles are shown as overlays on a timeline visualization. Each cluster is shown horizontally and advances over time from the left to the right. Both LBD reviews make substantial connections between #1 semantic predication and #6 validating discovery. Given the recency of #1 semantic predication, the role of semantic predication is significant. Our own ongoing research also investigates the role of semantic predication in understanding uncertainties of scientific knowledge [22]. Sebastian et al.'s review reached further down to #8 biomarker discovery, which was not cited in Smalheiser's review.

## Comparing five individual datasets

In order to compare the coverage of individual datasets, we construct a baseline map based on the combined dataset. The base map is created with the same procedure that was applied to individual datasets. The base map is divided into clusters. As a measure of the specificity of a dataset, we calculate the K-L divergence between normalized GCS and LCS scores. A low K-L

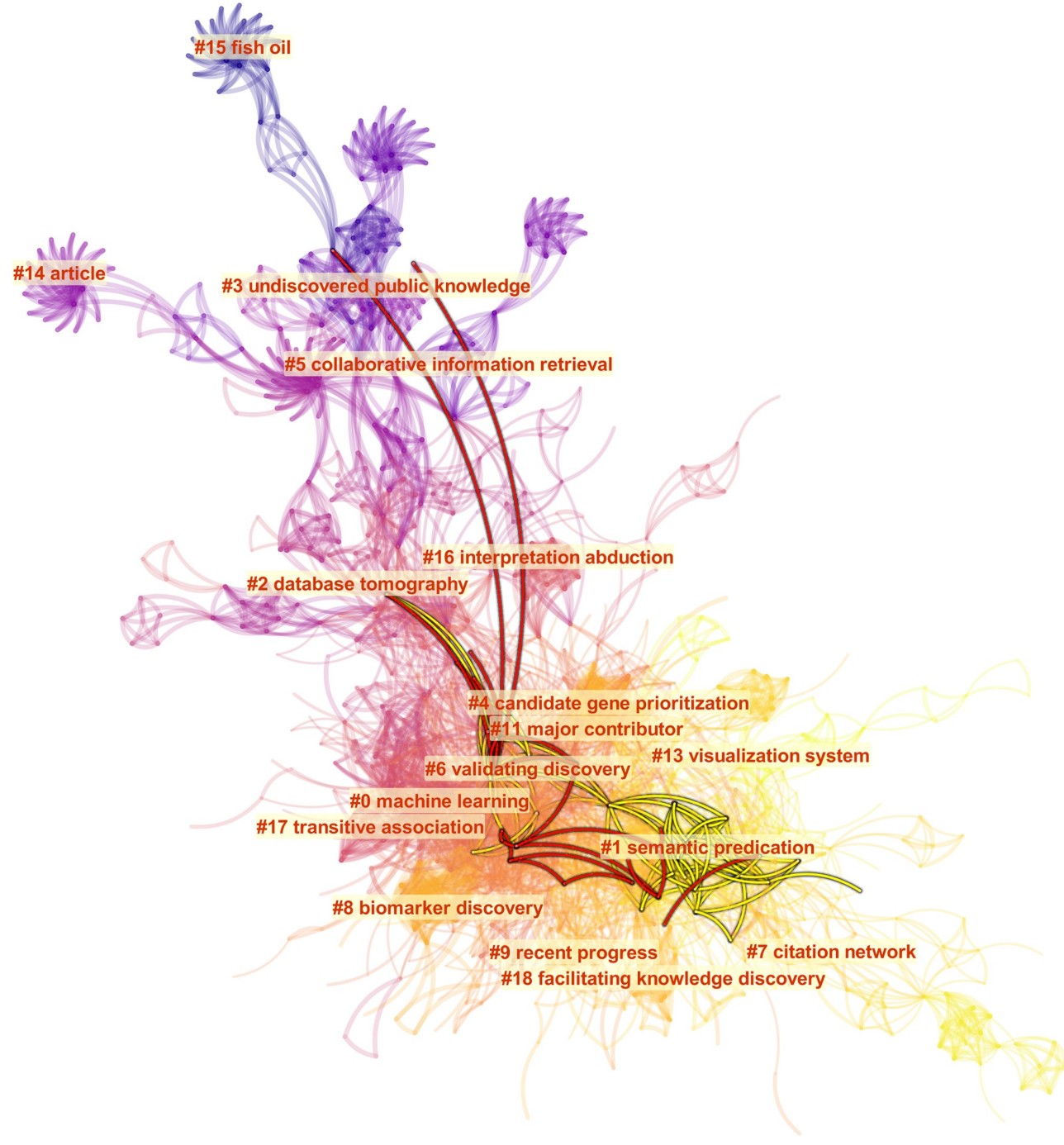

**Fig 4. A synthesized document co-citation network of the F dataset along with cluster labels and overlays of main paths of direct citations (red lines) and core references (yellow lines).** CiteSpace configuration: LRF = 3, LBY = 10, e = 2.0, g-index (k = 30). Network: 1,269 references and 5,937 co-citation links.

divergence would suggest that the dataset is representative of the underlying field of research, whereas a high K-L divergence would indicate that the dataset contains many out-of-place articles (Table 4). $S_3$ has the lowest K-L divergence. F and $N_F$ have similar scores. $N_B$ and $S_5$ have higher scores as expected given their size.

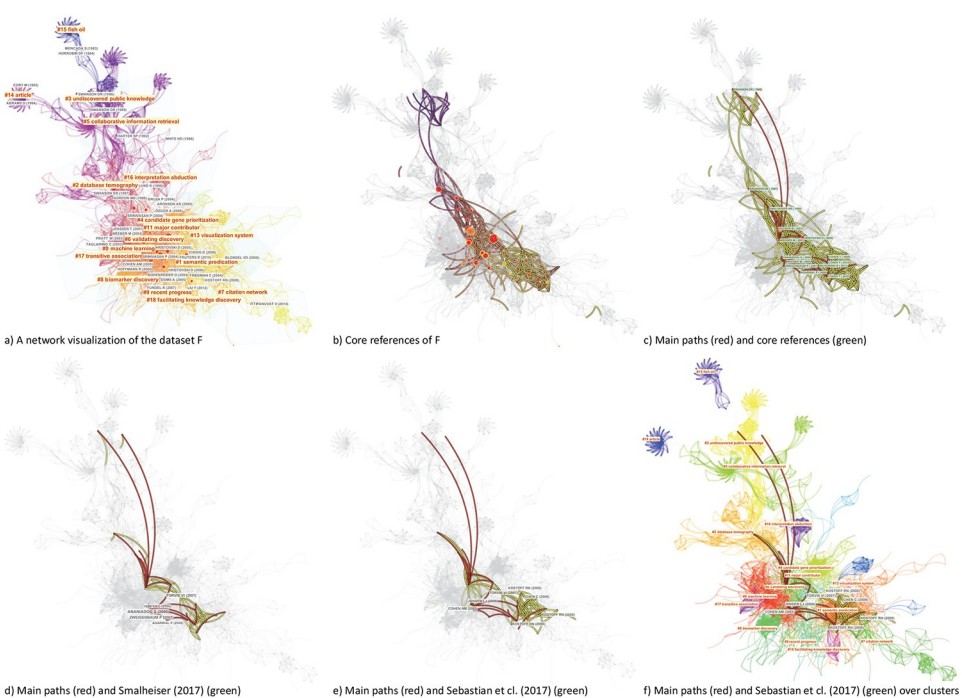

**Fig 5. A network visualization based on the dataset F and various overlays of substructures: a) a visualized network with clusters labeled, b) core references as an overlay, c) main paths overlay, d) an overlay of references cited by Smalheiser (2017), e) an overlay of references cited by Sebastian et al. (2017), and f) clusters are assigned distinct colors.**

The overview of a visualized network based on the combined dataset is shown in Fig 7. The colors in the map on the left depict the time of a link is added. For example, the youngest areas are located towards the lower left of the network, whereas the oldest ones are located near the top. The colors in the map on the right are encoded to depict the membership of clusters. The largest cluster is shown in red, following a rainbow colormap, so that we will know the relative size of a cluster. We will overlay networks from each individual dataset to identify what each search strategy brings unique topics to the overall landscape.

Table 5 lists the distribution of each of the largest 10 clusters in the combined network across the five individual datasets. Thematic labels of each cluster include terms selected by Latent Semantic Indexing (LSI) and by log-likelihood ratio. The former tends to identify common themes, whereas the latter tends to highlight unique themes. The two selections may differ as well as agree. Among the 10 largest clusters, the oldest one is #4 information retrieval with 1990 as the average year of publication. The youngest one is #8 deep learning with 2014 as the average year of publication. The largest cluster #0 systems biology/protein interaction network has the lowest silhouette score, which is expected given its size of 284 references. #6 microRNAs/drosophila melanogaster development has the highest silhouette score of 0.967, followed by the 0.965 of #7 big data, suggesting both them are highly uniformed.

The distributions of clusters across individual datasets show that clusters 0–1 and 3–5 are well represented in F with over 50% of the members of these cluster (highlighted in the table). $N_B$ is essentially responsible for Cluster #6, whereas $S_5$ is responsible for #7 big data and #8 deep learning. Independently we can identify the unique contributions associated with #6, #7, and #8 from network overlays shown in Fig 8.

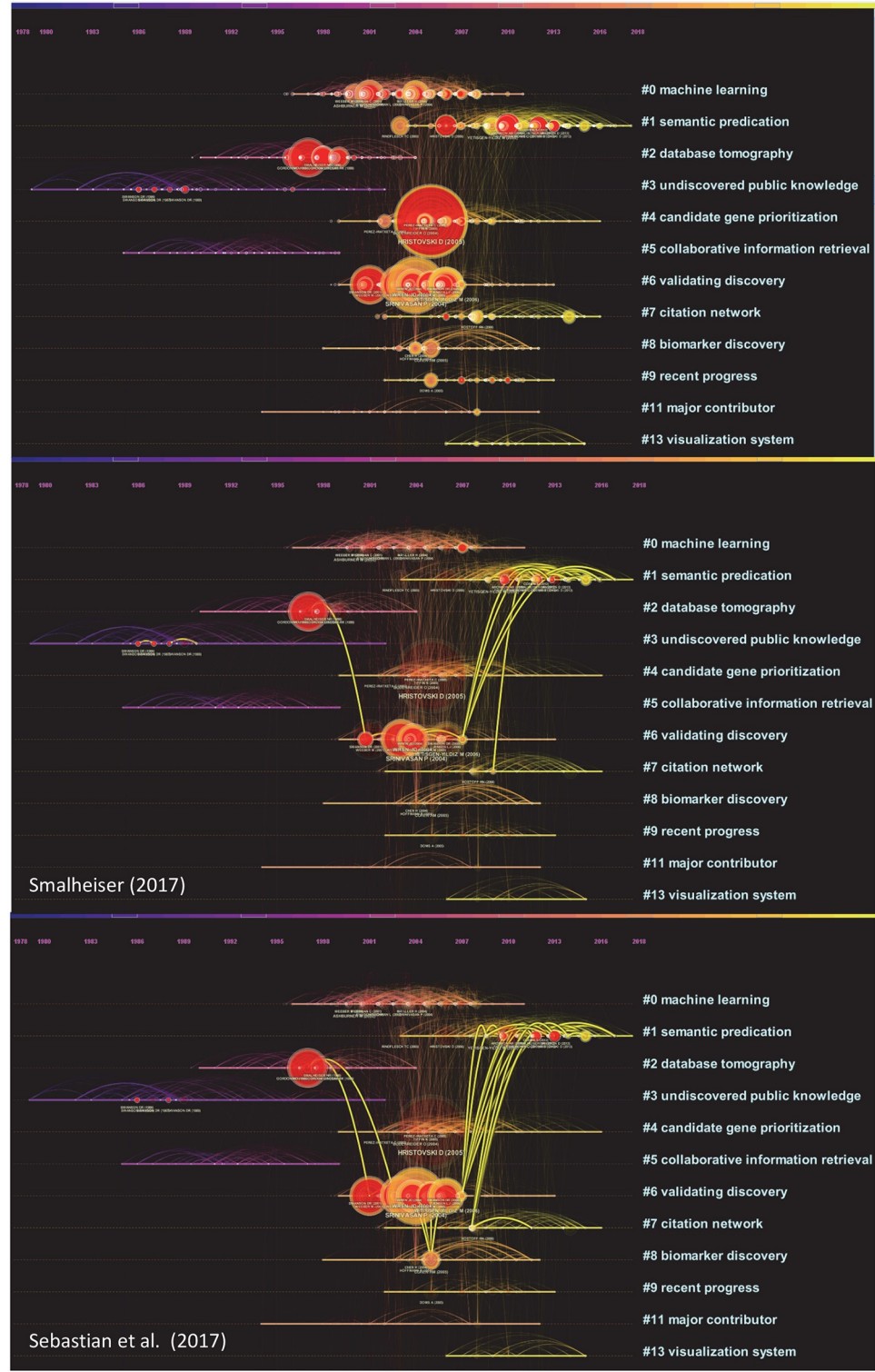

**Fig 6. Timeline visualization with overlays of references cited by two recent reviews of LBD, namely Smalheiser [28] and Sebastian et al. [5].**

**Table 4. The K-L divergences of the datasets and their cores.**

| Dataset | Range | max GCS | GCS≥3 | (%) | KL (GCS,LCS) | Core |
|---|---|---|---|---|---|---|
| F | 1983–2019 | 667 | 823 | 45.47 | 0.138 | 350 |
| $N_F$ | 1986–2018 | 3,919 | 68 | 93.15 | 0.135 | 59 |
| $N_B$ | 1936–2018 | 49,363 | 2,414 | 98.61 | 0.182 | 1,686 |
| $S_3$ | 1975–2018 | 4,969 | 748 | 100.00 | 0.070 | 86 |
| $S_5$ | 1975–2019 | 6,929 | 31,376 | 69.91 | 0.283 | 13,202 |
| All5 | 1936–2019 | 49,363 | 33,836 | 70.11 | 0.311 | 15,226 |

Fig 8 shows a set of network overlays of individual datasets (in red) and core reference overlays of the F and $S_5$ datasets. The three circles in part g highlight the three unique clusters. #6 is contributed essentially by $N_B$, whereas #7 and #8 are captured by $S_5$. The effect of cascading citation expansions is evident. The query-based approach (F) failed to capture #7 big data and #8 deep learning as the 5-generation forward expansion from Swanson's pioneering article did. Are these clusters relevant enough to be still considered as part of a systematic scientometric review of LBD or rather they should be considered as applications of computational technologies to literature-based discovery? Similarly, #6 microRNAs is missed by forward expansions from Swanson's 1986 article. What is the basis of its relevance? We will address these questions as follows.

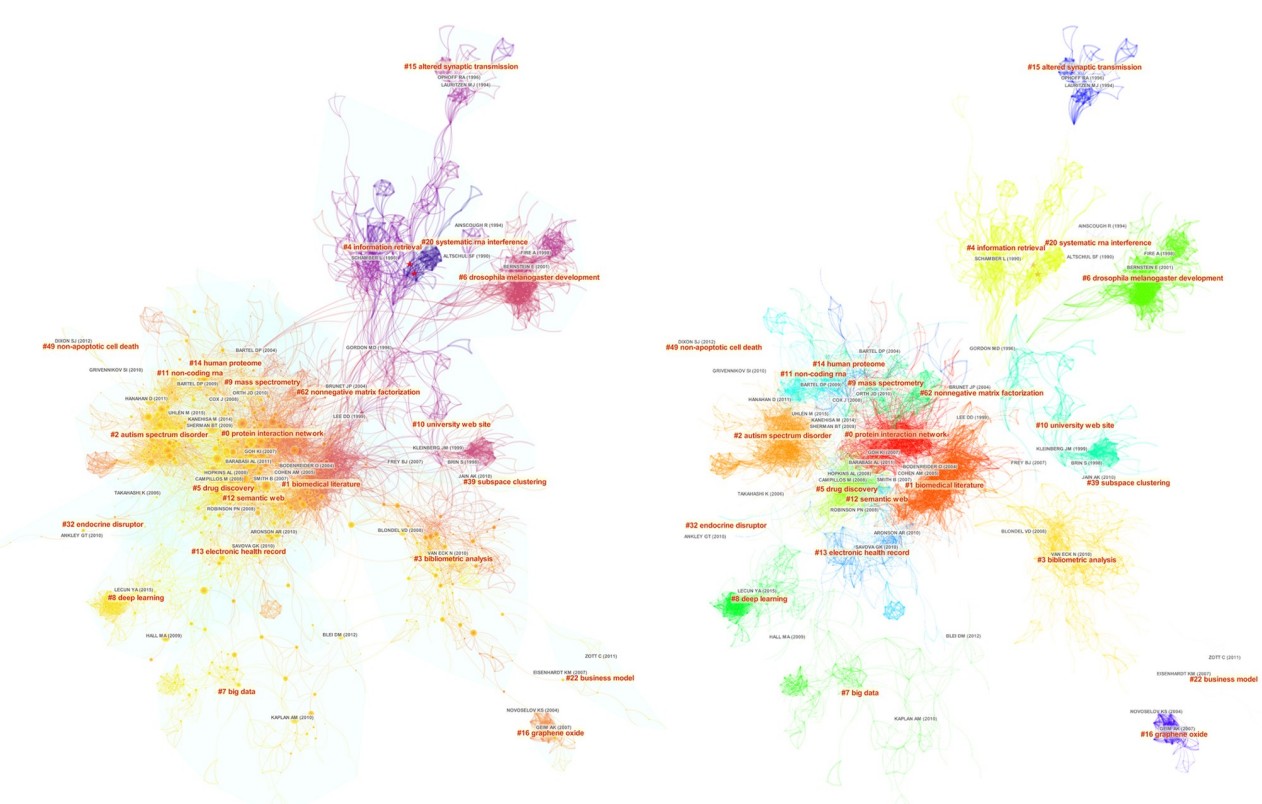

**Fig 7. A network visualization based on the combined dataset, featuring 3,095 references and 16,314 co-citation links.** Modularity: 0.84. Silhouette: 0.34.

**Table 5. Distributions of clusters across individual datasets.**

| Clusters | Major Themes of Citing Articles | Cluster Size | F | $N_F$ | $N_B$ | $S_3$ | $S_5$ |
|---|---|---|---|---|---|---|---|
| 0 | systems biology / protein interaction network | 283 | 70.0 | 4.9 | 73.9 | 65.0 | 100.0 |
| 1 | biomedical literature | 250 | 98.4 | 42.4 | 95.2 | 42.8 | 98.4 |
| 2 | cancer / autism spectrum disorder | 245 | 31.0 | 0.4 | 27.3 | 52.2 | 100.0 |
| 3 | science / bibliometric analysis | 214 | 69.6 | 13.6 | 59.3 | 29.9 | 100.0 |
| 4 | migraine / information retrieval | 183 | 54.6 | 13.7 | 51.9 | 12.6 | 84.2 |
| 5 | drug / drug discovery | 160 | 71.9 | 11.9 | 48.8 | 40.0 | 100.0 |
| 6 | microRNAs / drosophila melanogaster development | 143 | 4.9 | 23.8 | 100.0 | 8.4 | 67.8 |
| 7 | big data | 132 | 18.2 | 4.5 | 14.4 | 43.9 | 96.2 |
| 8 | deep learning | 117 | 10.3 | 4.3 | 15.4 | 27.4 | 99.1 |
| 9 | metabolomics / mass spectrometry | 106 | 24.5 | 1.9 | 25.5 | 56.6 | 100.0 |

Each cluster can be further analyzed by applying the same visual analytic procedure at the next level, i.e. Level 2. The cluster at the original level is known as a Level-1 cluster. One may continue this drill-down process iteratively as needed. Level-2 clusters are useful for interpreting their Level-1 cluster in terms of more specific topics.

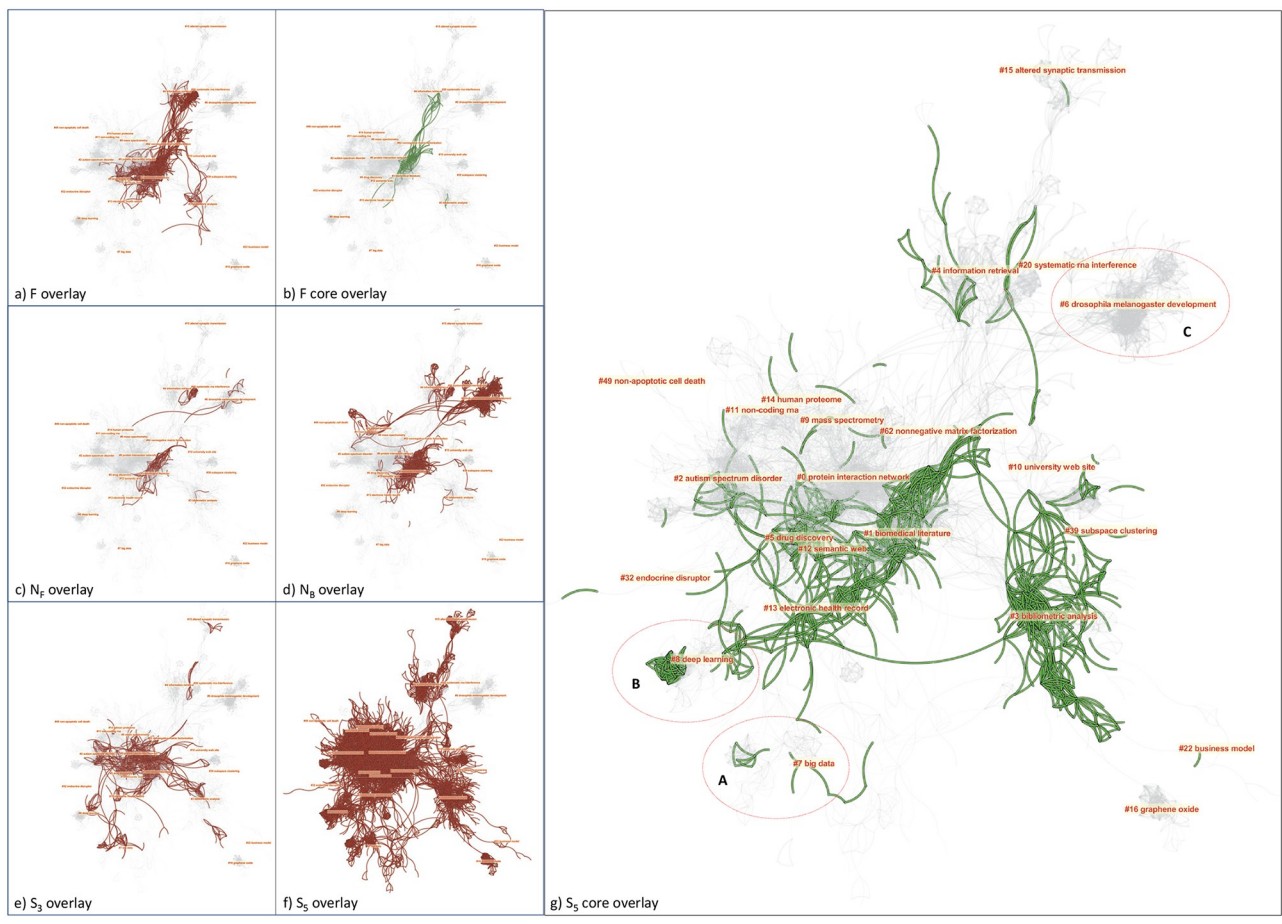

**Fig 8. Network overlays of individual datasets (in red) and core reference overlays (in green).**

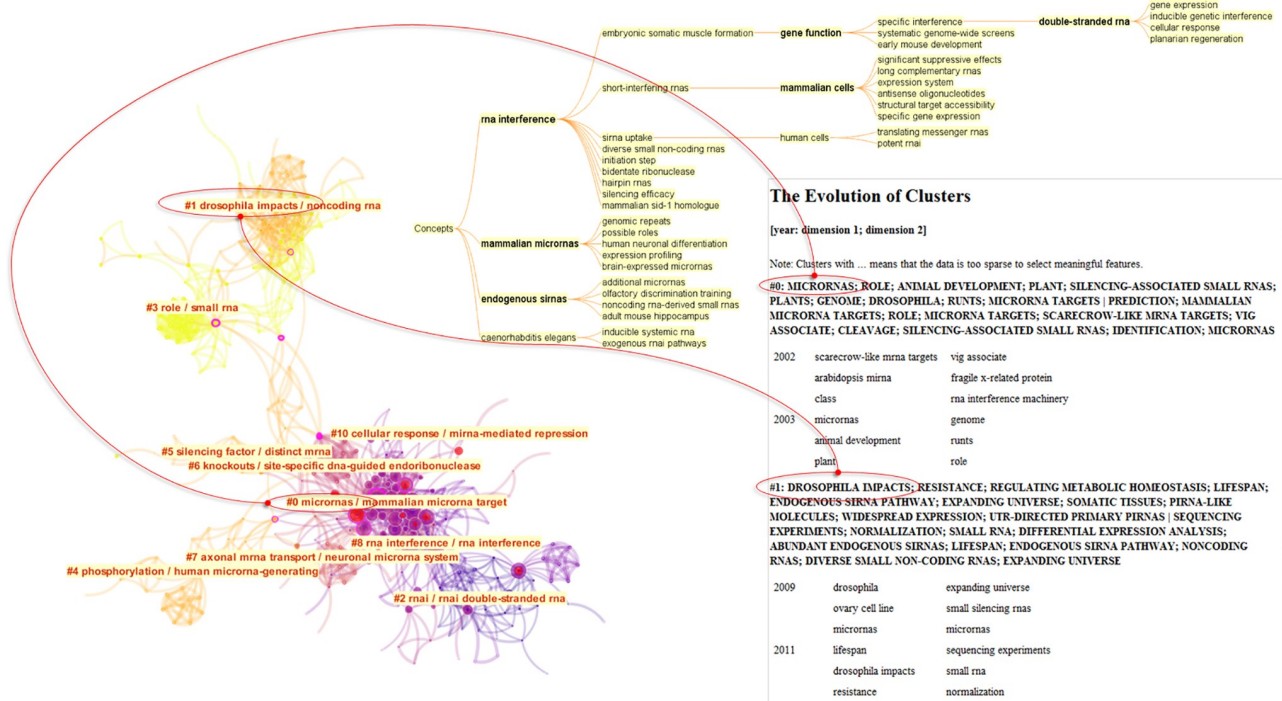

**Fig 9. Cluster #6 is primarily contributed by dataset N_B.** It is a very specific domain on RNA interference.

Fig 9 illustrates a few reports from CiteSpace on Cluster #, including a visualization that shows Level-2 clusters of the Level-1 cluster (left), a hierarchy of concepts (top), and year-by-year thematic terms of Level-2 clusters. The hierarchy of concepts, also known as a concept tree, provides a useful context to identify the major themes of a cluster according to the degree of a concept node in the tree, or the number of children in the tree. In this case, RNA interference has the highest degree and it suggests that the cluster's overarching theme is to do with RNA interference. A concept tree provides an informative context for selecting thematic labels. Labels selected through LSI or LLR do not have the benefit of such contextual information. As shown in Fig 9, #6 appears to be different from other clusters because connections to the study of scientific literature are not obvious.

Since N_B is exclusively responsible for the cluster #6, we overlay references cited by Smalheiser's review, i.e. its footprints, on a network visualization of N_B (Fig 10). The NB network consists of two components that are loosely connected with each other. The footprints of Smalheiser's review mostly appear in the lower component, indicating that the lower component is strongly relevant to LBD. In contrast, the upper component only contains two footprint references, namely Smalheiser NR (2001) and Lugli G (2005). The two references may hold the key to the formation of #6.

We examine the full text of Smalheiser's review for the contexts in which these two references are cited. As it turns out, Smalheiser cited the two references as atypical examples of LBD that "arose haphazardly during the course of laboratory investigation" and they are unlike typical LBD examples, in which complementary bodies of literature were purposefully sought after. Lugli G (2005) was cited in the first example of how Smalheiser and his colleagues put two lines of studies together that involved concepts such as double-stranded RNA, which is featured in the concept tree of #6 in Fig 9. Smalheiser NR (2001) was cited in the second

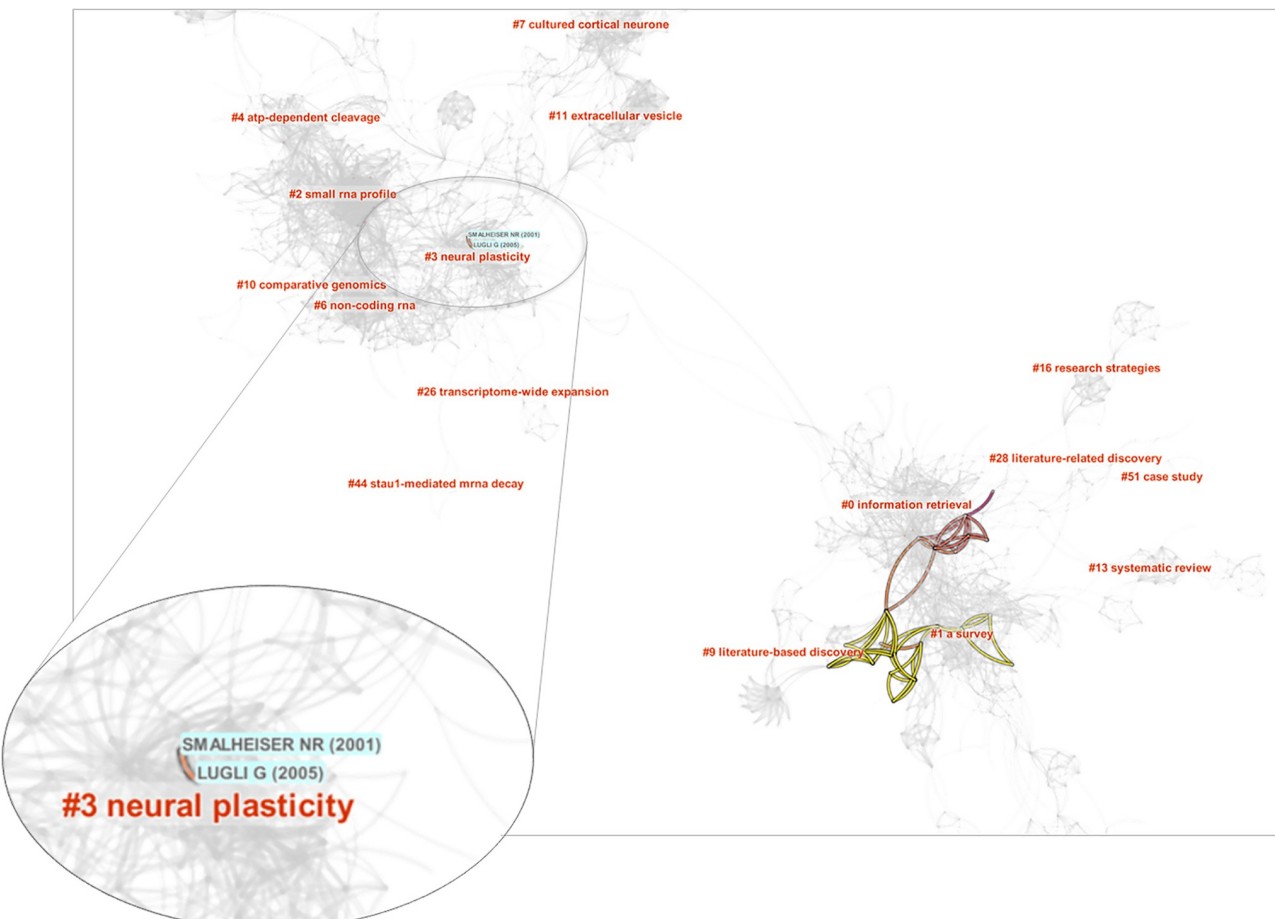

**Fig 10. The network of $N_B$ reveals two weakly connected continents.** The upper area is responsible for the formation of Cluster #6, which is in turn due to two references cited in Smalheiser in his 2017 review.

example of atypical LBD, which was about RNA interference in mammalian brain. It took them a decade to find provisional evidence that may valid the discovery in 2012.

Now it becomes evident that the upper component is connected to LBD through this specific connection. Thus the inclusion of #6 by expansion is reasonable. On the other hand, if the upper component does not contain any other references specifically relevant to LBD, then it seems to be necessary to investigate whether the $N_B$ expansion should be cut short in this area.

The query-based search (F) did not capture clusters #7 big data and #8 deep learning. As shown in Fig 11, the red lines indicate the coverage of F. No red lines even remotely approach to either of the clusters.

The relevance of Cluster #8 deep learning is investigated as follows. Fig 12 depicts a drill-down analysis of Cluster #8 deep learning, which is the youngest cluster among the 10 largest Level-1 clusters. The concept tree of the cluster identifies deep learning as the primary theme. More specifically, the concept of deep learning appears in contexts that are relevant to LBD, namely in association with drug discovery and biomedical literature. Level-2 clusters include #0 deep learning, #1 deep learning, #2 ensemble gene selection, #3 neuromorphic computing, #4 drug discovery, and #5 medical record. Year-by-year thematic terms include deep learning for the last four years since 2016 along with domain-specific terms such as radiology, breast ultrasound, and precision medicine. Given the multiple connections to biomedical literature,

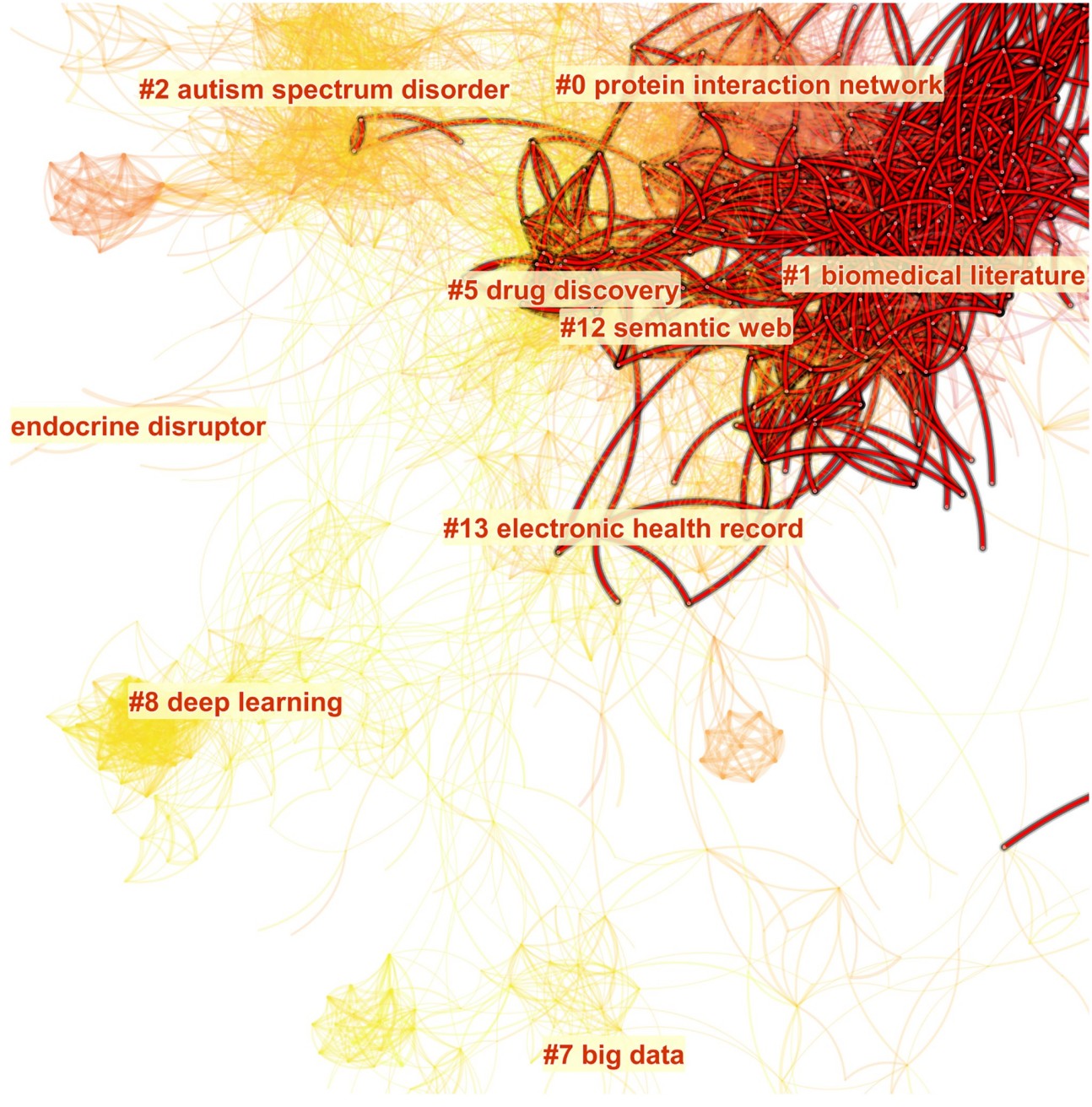

**Fig 11. Clusters #7 big data and #8 deep learning are captured by $S_5$ but not by the query-based search (F).**

drug discovery, and other domain-specific terms, the cluster on deep learning should be considered as a relevant development of LBD.

## Discussions and conclusions

We have proposed and demonstrated a flexible method to improve the quality of data retrieved for systematic scientometric reviews. We have demonstrated how one may use the approach to develop search strategies to meet the needs in common scenarios in practice. The comparisons

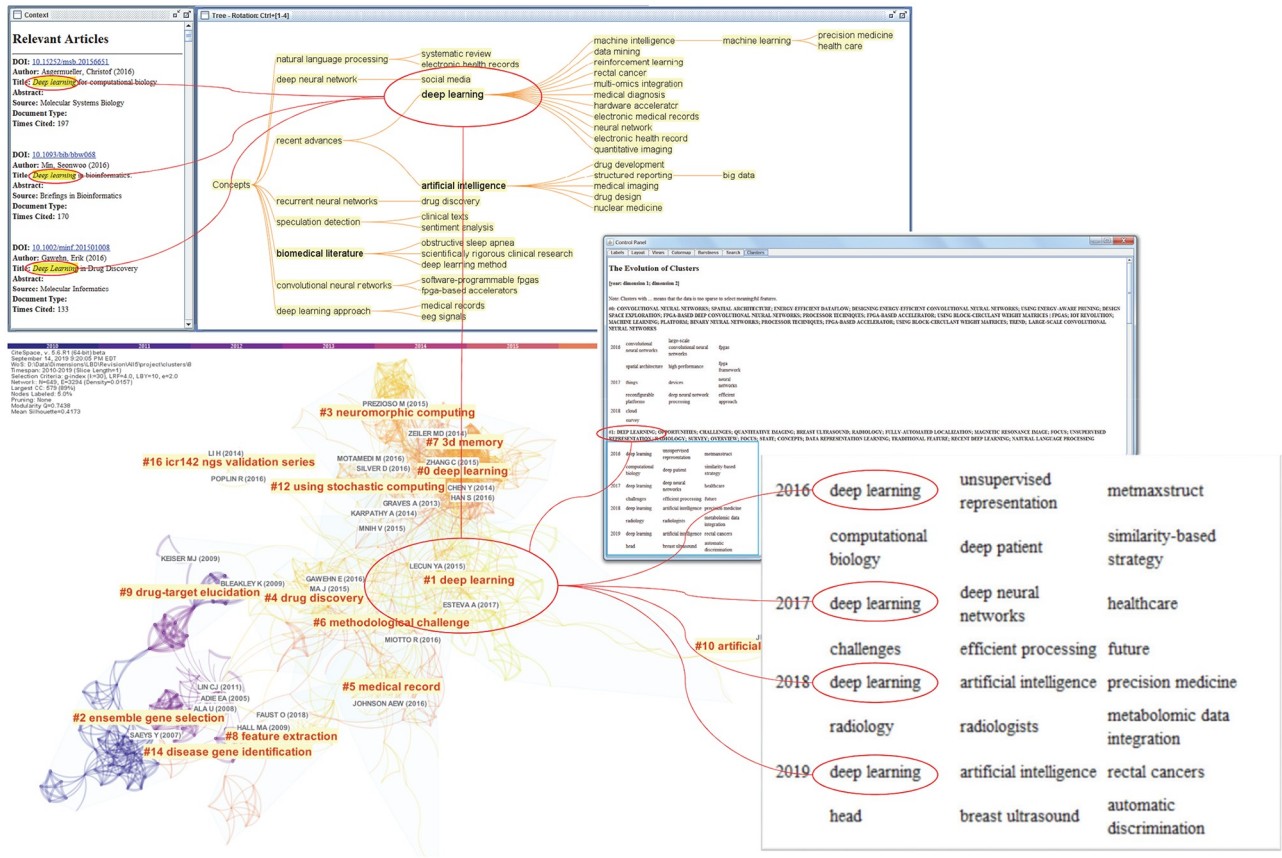

**Fig 12. Major Level-2 clusters of Level-1 cluster #8 deep learning.**

of network visualization overlays of five datasets have revealed what a commonly used full text search strategy could have missed. Such omissions are likely to be recently emerged topics and missing them in a systematic review may undermine its overall quality. A strategy that combines query-based search and cascading citation expansion is likely to provide a more balanced coverage of a research domain and to reduce the risk of overly relying on topics explicitly specified in the initial queries. A practical implication on finding a representative body of the literature is its potential to uncover emerging topics that are currently connected to the main body of the literature through a chain of weak links. We recommend researchers to consider this strategy in situations when they only have a small number of relevant articles to begin with. As our study demonstrated, a wide variety of articles can serve as a starting point of an expansion process and multiple processes can be utilized and the combination of their results is likely to provide a comprehensive coverage of the underlying thematic landscape of a research field or a discipline.

The present study has some limitations and it raises new questions that need to be addressed in future studies. Our approach implies an assumption that the structure of scientific knowledge can be essentially captured through semantically similar text and/or explicit citation links. Is this assumption valid at the disciplinary level? To what extent does the choice of the seed articles for the expansion process matter? Does the choice of seed articles influence the stability of the expansion process? How many generations of expansion would be optimal?

We have made a few observations and recommendations that are potentially valuable for adapting this type of search strategies to develop a systematic review of a body of scientific literature of interest.

- Using a combination of multiple cascading citation expansions with different seed articles is recommended to obtain a more balanced representation of a field than using a full text search alone.

- Multi-generation citation expansions provide a systematic approach to reduce the risk of missing topics that we may not be familiar with or not aware of altogether.

- Triangulating multiple aggregations of articles such as the core references of a dataset and main paths of a dataset as well as multiple review articles provide useful insights.

- The flexibility of the approach enables researchers to apply the expansion and visual analytic procedure iteratively at multiple levels of granularity, for example, expanding a cluster, comparing the footprints of review articles with main paths, and drilling down a cluster in terms of Level-2 clusters.

- Choosing the starting point and an end point of a cascading expansion process may lead to different results, suggesting the complexity of the networks and threshold selections may play important roles in reproducing the results in similar studies.

- Modularity and cluster silhouette measures can help us to assess the quality of an expansion process.

Comparing multiple networks in the same context allows us to identify the topic areas that are particularly well represented in some of the datasets but not in other ones. Such an understanding of the landscape of a field provides additional insights into the structure and the long-term development of the field.

As a methodology for generating systematic scientometric reviews of a knowledge domain, it bridges the formally mutual exclusive globalism and localism by providing a scalable transition mechanism between them. The most practical contribution of our work is the development and dissemination of a tool that is readily accessible by end users.

## Acknowledgments

We are grateful to Neil Smalheiser for his valuable suggestions on the paper. The work is supported by the SciSIP Program of the National Science Foundation (Award #1633286). CC acknowledges the support of Microsoft Azure Sponsorship. Data sourced from Dimensions, an inter-linked research information system provided by Digital Science (https://www. dimensions.ai). This work is also supported by the Ministry of Education of the Republic of Korea and the National Research Foundation of Korea (NRF-2018S1A3A2075114). This research is also partially supported by the Yonsei University Research Fund of 2019-22-0066.

## Author Contributions

**Conceptualization:** Chaomei Chen, Min Song.

**Data curation:** Chaomei Chen, Min Song.

**Formal analysis:** Chaomei Chen, Min Song.

**Funding acquisition:** Chaomei Chen.

**Methodology:** Chaomei Chen, Min Song.

**Software:** Chaomei Chen.

**Visualization:** Chaomei Chen.

**Writing – original draft:** Chaomei Chen, Min Song.

**Writing – review & editing:** Chaomei Chen, Min Song.

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
