## [Decision Letter · Decision Letter 0]

2 Aug 2019

PONE-D-19-16569

Visualizing a Field of Research: A Methodology of Systematic Scientometric Reviews

PLOS ONE

Dear Dr. Chen,

Thank you for submitting your manuscript to PLOS ONE. After careful consideration, we feel that it has merit but does not fully meet PLOS ONE’s publication criteria as it currently stands. Therefore, we invite you to submit a revised version of the manuscript that addresses the points raised during the review process.

I would follow Reviewer 2, who points to several majors issues that can be linked to two fundamental problems. In particular, the promised simplification of the conceptualisation of globalism and localism is not convincingly tackled and to a certain extent, the manuscript lacks proper documentation of methodology and its application. I would like to ask the authors to remedy these issues and also to resolve the minor issues according to the reviewer’s comments.

We would appreciate receiving your revised manuscript by Sep 16 2019 11:59PM. To enhance the reproducibility of your results, we recommend that if applicable you deposit your laboratory protocols in protocols.io, where a protocol can be assigned its own identifier (DOI) such that it can be cited independently in the future. For instructions see: http://journals.plos.org/plosone/s/submission-guidelines#loc-laboratory-protocols

We look forward to receiving your revised manuscript.

Kind regards,

Wolfgang Glanzel, PhD

Academic Editor

PLOS ONE

Journal Requirements:

2.  We note that Figure 2 in your submission contains a copyrighted image. All PLOS content is published under the Creative Commons Attribution License (CC BY 4.0), which means that the manuscript, images, and Supporting Information files will be freely available online, and any third party is permitted to access, download, copy, distribute, and use these materials in any way, even commercially, with proper attribution. For more information, see our copyright guidelines: http://journals.plos.org/plosone/s/licenses-and-copyright.

1.    You may seek permission from the original copyright holder of Figure(s) [2] to publish the content specifically under the CC BY 4.0 license.

3. Please upload a new copy of Figures 4 - 17 as the detail is not clear. Please follow the link for more information: http://blogs.PLOS.org/everyone/2011/05/10/how-to-check-your-manuscript-image-quality-in-editorial-manager/

Reviewers' comments:

Reviewer's Responses to Questions

**Comments to the Author**

1. Is the manuscript technically sound, and do the data support the conclusions?

Reviewer #1: Yes

Reviewer #2: Partly

2. Has the statistical analysis been performed appropriately and rigorously? 

Reviewer #1: Yes

Reviewer #2: N/A

3. Have the authors made all data underlying the findings in their manuscript fully available?

Reviewer #1: Yes

Reviewer #2: Yes

4. Is the manuscript presented in an intelligible fashion and written in standard English?

Reviewer #1: Yes

Reviewer #2: Yes

5. Review Comments to the Author

Reviewer #1: The authors have demonstrated how to use cascading citation expansion method to generally increase the body of literature to be included and analyzed. The integration between Dimesions and CiteSpace is beneficial to bibliometricians. The method described in the current manuscript offers an alternative solution to define a research landscape, which can be used in parallel with other bibliometric approaches such as cited reference analysis with CRExplorer.

Reviewer #2: This paper compares different ways of creating scientometric datasets in an attempt to identify a method that maximizes coverage of relevant documents while minimizing extraneous material.

I have several major concerns with the paper.

First, the paper is framed in terms of aiding systematic scientometric reviews and in bridging the local-to-global continuum in terms of datasets. However, the paper does little to address either of these issues. In fact, there is little to no referencing of these framing features throughout the results and discussion other than a simple reprisal of the claims at the very end of the paper. So, while the issues used to frame the paper are real, the body of the paper really doesn’t address the issues. I would thus suggest that the paper be framed more simply in terms of differences between datasets, which is what the paper is really about.

Second, and perhaps more importantly, the purpose of the paper is to contrast various ways of creating a dataset. However, the analysis is a full step removed from this because the results of each query (and the superset) are not directly compared. Instead, these results are run through the CiteSpace co-citation black box which hugely amplifies some queries while dramatically reducing the F5 query and the overall space, in addition to being a citation generation removed from all of the queries. Co-citation does tend to remove the cutting edge of research fronts, and since reviews are often focused on the hot or emerging topics, I fail to see how this method could be a great basis for aiding in reviews. I’m not saying that CiteSpace/co-citation should not be used, but I am saying that the authors need to honestly and accurately characterize what they are really doing. This does not currently come through in the paper.

Related to this, the CiteSpace step (bridging Tables 3 and 4) is all of one or two sentences, and the first impression from comparing the numbers of articles/nodes in Tables 3 and 4 is one of bewilderment. What is going on here. How does NF go from 73 articles to 1903 nodes. Also, why does CiteSpace end up with a much smaller set for F5 and smaller sets for the other queries. This all seems to be magic, and much more detail is required for this transition.

Finally, the comparison is qualitative, which it has to be given the lack of ground truth. On the other hand, the authors chose this subject because of their familiarity with it, and yet fail to mention which of the clusters are within vs. without what they would consider LBD to encompass. They seem to favor the F5 expansion. Are there clusters in the F5 that, because of over-expansion, are really outside the field? We are left wanting a more definitive result.

Minor issues:

Michel Zitt wrote about citation expansion several times, most directly in IPM 2006, but also presaged in Scientometrics 1996 and 2003. Many people have used citation expansion – for instance the original Places&Spaces display had a three-generation backward citation expansion based on Nobel Prize winners and others that was displayed on a science map. Author Chen also has a 2006 paper with forward expansion. So this paper is not introducing citation expansion, but is definitely refining it. Claims in this paper should be adjusted accordingly.

Although creation of global maps is indeed “limited to a small number of researchers”, it is also true that a global model is now available to all users of SciVal, which has over a thousand institutional subscribers, and is being widely used for institutional portfolio analysis. While the detailed information in this global model is not used widely to seed systematic reviews, it could be.

The authors claim that local maps are free from the need for stability. I disagree. If there is any comparison to be done from one point in time to another, stability is still needed, and cannot be achieved by a local map.

When it comes to query-based search, I would suggest using a reference to at least one paper by Alan Porter that contains a detailed multi-part query. Also, to be fair to the detailed multi-part query, these queries are typically designed in a multi-step process. A query is run, results are examined to see what should be dropped and/or added, and this is done multiple times, which results in the detailed query with presumable high precision/recall. This should be mentioned. Porter’s paper will mention this process.

Figure 3 would be far easier to interpret if the y-axis used base10 for its log scale.

In the Figure 7 overlay maps, it is very difficult to tell what is overlay and what is basemap. It would be far easier to distinguish the overlay if the basemap were in gray or some other neutral color, and the overlay were colored.

6. PLOS authors have the option to publish the peer review history of their article (what does this mean?). If published, this will include your full peer review and any attached files.

Reviewer #1: No

Reviewer #2: No

---

## [Author Response · Author response to Decision Letter 0]

16 Sep 2019

PONE-D-19-16569

Visualizing a Field of Research: A Methodology of Systematic Scientometric Reviews

PLOS ONE

Dear Editor,

First, we would like to thank the reviewers for providing their comments. 

We have carefully revised the manuscript based on the reviews. Most of the manuscript has been re-written to improve the clarity of the description. Most of the figures are re-generated as well.

We summarize our responses to the reviewers as follows. The text reflects the changes accordingly.

Best wishes,

Chaomei Chen and Min Song

2. We note that Figure 2 in your submission contains a copyrighted image. All PLOS content is published under the Creative Commons Attribution License (CC BY 4.0), which means that the manuscript, images, and Supporting Information files will be freely available online, and any third party is permitted to access, download, copy, distribute, and use these materials in any way, even commercially, with proper attribution. For more information, see our copyright guidelines: http://journals.plos.org/plosone/s/licenses-and-copyright.

Response: Figure 2 was in fact a screenshot of an interface in CiteSpace for connecting to Dimensions. Thus we would be its owner. Regardless, the figure is now replaced with a new Figure 2 to illustrate the workflow of handling the datasets and major analytic tasks.

Comments to the Author

5. Review Comments to the Author

Response: We are grateful to the reviewers for providing their comments. We have thoroughly revised the study to address these comments. We have in fact gone through the entire process and updated most of the figures and tables along with interpretations and discussions. As a result, the manuscript has been re-written substantially. Please be aware of the possible shifts in text between the revised version and the original version in terms of references to figures and details such as clusters. 

Reviewer #1: The authors have demonstrated how to use cascading citation expansion method to generally increase the body of literature to be included and analyzed. The integration between Dimesions and CiteSpace is beneficial to bibliometricians. The method described in the current manuscript offers an alternative solution to define a research landscape, which can be used in parallel with other bibliometric approaches such as cited reference analysis with CRExplorer.

Response: Thank you.

Reviewer #2: This paper compares different ways of creating scientometric datasets in an attempt to identify a method that maximizes coverage of relevant documents while minimizing extraneous material.

Response: Thank you. This is indeed a very accurate outline of the study.

I have several major concerns with the paper.

First, the paper is framed in terms of aiding systematic scientometric reviews and in bridging the local-to-global continuum in terms of datasets. However, the paper does little to address either of these issues. In fact, there is little to no referencing of these framing features throughout the results and discussion other than a simple reprisal of the claims at the very end of the paper. So, while the issues used to frame the paper are real, the body of the paper really doesn’t address the issues. I would thus suggest that the paper be framed more simply in terms of differences between datasets, which is what the paper is really about.

Response: While the paper indeed focuses on the differences between datasets, the motivation for doing so is driven by the lack of flexibility in access the local-global continuum. The cascading citation expansion presented in the paper offers a method that enables us to iteratively expand a set of articles so as to include relevant articles that are not matched by initial search queries. Therefore we believe this paper offers a concrete method in response to the local-global dilemma when researchers try to find publications with high recall (in theory in favor of global approaches) and up-to-date structures (as global structures may not be updated as frequently as needed). As we show in the paper, with cascading citation expansions in different combinations, the boundaries of the topic of interest are no longer limited by initial search. 

Another advantage of citation-based expansion is to reduce the burden of having to rely on domain experts in constructing complex queries and thus to shift the focus of domain experts from the question of what we may miss to the question of whether a topic in front of us is relevant enough. 

The local-global continuum provides a necessary context of the study also in connection to the principles of literature-based discovery, namely to discovery new connections, including individual domain experts may not be aware of, as one cannot expect to have access to domain experts who know everything about the research area to be surveyed.

We have revised the text with reference to the role of the local-global continuum.

Second, and perhaps more importantly, the purpose of the paper is to contrast various ways of creating a dataset. However, the analysis is a full step removed from this because the results of each query (and the superset) are not directly compared. Instead, these results are run through the CiteSpace co-citation black box which hugely amplifies some queries while dramatically reducing the F5 query and the overall space, in addition to being a citation generation removed from all of the queries. Co-citation does tend to remove the cutting edge of research fronts, and since reviews are often focused on the hot or emerging topics, I fail to see how this method could be a great basis for aiding in reviews. I’m not saying that CiteSpace/co-citation should not be used, but I am saying that the authors need to honestly and accurately characterize what they are really doing. This does not currently come through in the paper.

Response: Thank you for your comments. We added direct comparisons between different datasets with new tables (Tables 3, 4). In general, they capture many articles that are not in the F set, i.e. the query search result. For example, the S5 set contains 44,883 articles with only 702 of them found in F. More details can be found in the manuscript.

There are a few reasons for using CiteSpace. First, the original motivation of the study is to improve the quality of systematic scientometric studies supported by CiteSpace and a few other tools such as VOSviewer. Improving the quality of input data is a common need for applications of scientometric tools. Second, CiteSpace allows us to inspect the interrelationships between datasets obtained from different expansions in context so as to contrast which areas are covered well by particular expansions and which areas are underrepresented. The visualization process retains articles with at least 2 citations in at least one year. The resultant map does not represent all the articles in a dataset, only those with sufficient citations. This is a justifiable selection and it is used widely in practice, including global approaches.

Co-citation maps in CiteSpace have grown out of the classic notion of co-citation considerably by incorporating information from citing articles to a great extent. Perhaps we ought to give it a new name to indicate the difference instead of keeping calling it a co-citation network. Notably, clusters shown in the interactive visualization represent a duality between citing articles and cited references. Cluster labels are noun phrases extracted from citing articles to cited references in corresponding clusters. Year-by-Year labels of a cluster provide detailed information on emerging topics of citing articles. For example, clusters such as big data and deep learning are identified in the study. Citing articles to these clusters feature many articles as recent as 2018, considering the data was collected in early 2019. CiteSpace does support functions for generating maps based on bibliographic coupling. In this study, we limit our focus to how cascading citation expansion may affect subsequently generated science maps. We believe citation/co-citation provides an additional vetted information to the bibliometric landscape that alternative methods may not readily reveal. Note that there are a wide variety of possible routes to conduct a visual analytic study of a given dataset. Here these examples are primarily provided to demonstrate the major differences of different strategies of data collection.

As a side note, the ‘black-box’ process in CiteSpace has many user-controllable parameters for users to verify the role of each parameter in the workflow, although the complexity involved in the process is indeed non-trivial in terms of the number of decisions to make when selecting a particular workflow to proceed.

We update the text accordingly to clarify the above points and to provide more detailed descriptions of the steps. 

Related to this, the CiteSpace step (bridging Tables 3 and 4) is all of one or two sentences, and the first impression from comparing the numbers of articles/nodes in Tables 3 and 4 is one of bewilderment. What is going on here. How does NF go from 73 articles to 1903 nodes. Also, why does CiteSpace end up with a much smaller set for F5 and smaller sets for the other queries. This all seems to be magic, and much more detail is required for this transition.

Response: We have revised the descriptions to make them self-contained and added references to publications that describe the relevant information in further detail. The two tables are updated as well.

The Nf dataset contains 73 articles, or source articles, or citing articles. The 73 articles cite 2,239 unique references. 1,903 of them satisfied the selection criteria based on local citation counts and they become the nodes in the corresponding network of references for Nf. 

Similarly, S5 contains a much larger number of citing articles, which in turn cited an even larger number of references. References must meet the selection criteria to be retained. As shown in publications such as van Noorden et al. (2014) the majority of the paper mountain has papers with zero citations globally. 

We have revised the text accordingly to clarify the details.

Finally, the comparison is qualitative, which it has to be given the lack of ground truth. On the other hand, the authors chose this subject because of their familiarity with it, and yet fail to mention which of the clusters are within vs. without what they would consider LBD to encompass. They seem to favor the F5 expansion. Are there clusters in the F5 that, because of over-expansion, are really outside the field? We are left wanting a more definitive result.

Response: We give a higher priority to reduce the risk of missing important relevant articles than including less relevant ones. In part, we share the view with the literature-based discovery in general in that we consider it is critical to bring otherwise disparate bodies of scholarly publications to the attention of researchers. Furthermore, the quality of the coverage directly impacts subsequent analytic tasks, including the assessment of relevance of a given set of articles retrieved. 

In the revised version, we particularly discuss the relevance issue with reference to clusters on big data, deep learning, and human metabolisms. The first two clusters are considered relevant with reference to the literature-based discovery research based on the roles they played in those studies (through inspections in cluster explorer), whereas the third one (cluster #6), with a specific focus on domain specific topics such as microRNA and RNA interference and without strong evidence of connections to literature-based discovery, is considered beyond the scope of literature-based discovery.

We have revised the text accordingly.

Minor issues:

Michel Zitt wrote about citation expansion several times, most directly in IPM 2006, but also presaged in Scientometrics 1996 and 2003. Many people have used citation expansion – for instance the original Places&Spaces display had a three-generation backward citation expansion based on Nobel Prize winners and others that was displayed on a science map. Author Chen also has a 2006 paper with forward expansion. So this paper is not introducing citation expansion, but is definitely refining it. Claims in this paper should be adjusted accordingly.

Response: Michael Zitt (2006) in IPM is indeed relevant and useful. We have added an overlay of the core references of a dataset based on the ratio of local- and global citations. The core list is useful as an additional level of aggregation of references to highlight the boundary of the core references. 

Theoretically, the concept of citation expansion is indeed conceivable from the original introduction of citation indexing. Pragmatically, the workflow has remained to be outside of the access of most researchers. We emphasize the practical contribution of the cascading part of the citation expansion, which would require a sustained access to the source database. Therefore, we welcome the new opportunities enabled by the Dimensions API. The 5-generation citation expansions illustrated in the paper may continue further and thus reduce the gap between a local- and a global view of the literature. More importantly to us, our work enables many researchers to adopt the workflow by applying the freely available functions supported in CiteSpace to the vast coverage of the Dimensions platform.

We have adjusted our claims in the revised text accordingly. 

Although creation of global maps is indeed “limited to a small number of researchers”, it is also true that a global model is now available to all users of SciVal, which has over a thousand institutional subscribers, and is being widely used for institutional portfolio analysis. While the detailed information in this global model is not used widely to seed systematic reviews, it could be.

Response: Yes, it could. Updating a global model is much more time/effort consuming than creating a local model from scratch. According to Borner et al. (2012), the 2005 USCD map was created in 2007. Its next version 2010 USCD map was reported in 2012, adding six years of data from WoS and 3 years from Scopus. The issue is how long the global maps can stay valid as the structure depicted by the existing model is constantly subject to the changes introduced by new publications and how sensitive the current model to changes that may present only at a finer level of granularity, e.g., as shown in local maps. What we contribute here is a compromise that offers an increased flexibility and agility.

The authors claim that local maps are free from the need for stability. I disagree. If there is any comparison to be done from one point in time to another, stability is still needed, and cannot be achieved by a local map.

Response: The sentence you referred to is probably this one: “Localized science maps are free from the need to maintain a stable structure.” What we intend to convey here is that for local maps the structure is constructed based on the current data, thus the most recent data is taken into account. In contrast, if a global map is created a few years ago, then it is a legitimate question whether any subsequently published articles might have altered the structure significantly enough, especially at finer levels of granularity. 

If we define global maps as the ones featuring all scientific disciplines (Borner et al. 2012), then it is certainly achievable by a local map to make comparisons. What we need in such scenarios is a sufficiently large context to accommodate snapshots taken at different time points. The Link Walkthrough function in CiteSpace allows users to step through the visualized network one time slice at a time so that users can see which areas are covered in a particular time slice. 

We have revised the text accordingly.

When it comes to query-based search, I would suggest using a reference to at least one paper by Alan Porter that contains a detailed multi-part query. Also, to be fair to the detailed multi-part query, these queries are typically designed in a multi-step process. A query is run, results are examined to see what should be dropped and/or added, and this is done multiple times, which results in the detailed query with presumable high precision/recall. This should be mentioned. Porter’s paper will mention this process.

Response: We have added the following papers to the related work/discussion, two by Porter and his coauthors and one by Kostoff et al. They are in the same category of constructing complex queries through iterative processes. The key strength of such approaches is the input of domain-specific expertise, which also makes it a potential weakness of the method if the domain expertise is not readily accessible. It is cognitively more demanding to think of what is missing in the queries than to review whether specific articles retrieved are indeed relevant. Besides, multi-part complex query approaches can also benefit from cascading citation expansion as the initial burden on the domain experts would be much reduced. 

In addition, in consistent with the basic idea of literature-based discovery, it is valuable to undiscover disparate bodies of relevant literature that even domain experts may not be aware of. In the IPM paper you recommended to us , Zitt identified some of the issues with query-based approaches: “The reasons for the absence are various: terms forgotten by experts; terms deliberately excluded because of the fear of noise generated by too general or ambiguous formulation; terms deliberately excluded as deemed out of the scope of the topic.” P. 1526.

• Huang Y, Schuehle J, Porter AL, Youtie J. A systematic method to create search strategies for emerging technologies based on the Web of Science: illustrated for ‘Big Data’. Scientometrics. 2015;105(3):2005-22. doi: 10.1007/s11192-015-1638-y.

• Porter AL, Youtie Y, Shapira P, Schoeneck DJ. Refining search terms for nanotechnology. Journal of Nanopartical Research. 2008;10(5):715-28. doi: 10.1007/s11051-007-9266-y.

• Kostoff RN, Koytcheff RG, Lau CGY. Technical structure of the global nanoscience and nanotechnology literature. Journal of Nanopartical Research. 2007;9(5):701-24.

Figure 3 would be far easier to interpret if the y-axis used base10 for its log scale.

Response: Thank you for your comments. Revised as such.

In the Figure 7 overlay maps, it is very difficult to tell what is overlay and what is basemap. It would be far easier to distinguish the overlay if the basemap were in gray or some other neutral color, and the overlay were colored.

 Response: Thank you for your comments. Revised as such.

 Thanks again for reviewers’ comments!

---

## [Decision Letter · Decision Letter 1]

3 Oct 2019

Visualizing a Field of Research: A Methodology of Systematic Scientometric Reviews

PONE-D-19-16569R1

Dear Dr. Chen,

We are pleased to inform you that your manuscript has been judged scientifically suitable for publication and will be formally accepted for publication once it complies with all outstanding technical requirements.

With kind regards,

Wolfgang Glanzel, PhD

Academic Editor

PLOS ONE

Additional Editor Comments (optional):

Reviewers' comments:

Reviewer's Responses to Questions

**Comments to the Author**

1. If the authors have adequately addressed your comments raised in a previous round of review and you feel that this manuscript is now acceptable for publication, you may indicate that here to bypass the “Comments to the Author” section, enter your conflict of interest statement in the “Confidential to Editor” section, and submit your "Accept" recommendation.

Reviewer #2: All comments have been addressed

2. Is the manuscript technically sound, and do the data support the conclusions?

Reviewer #2: Yes

3. Has the statistical analysis been performed appropriately and rigorously? 

Reviewer #2: N/A

4. Have the authors made all data underlying the findings in their manuscript fully available?

Reviewer #2: Yes

5. Is the manuscript presented in an intelligible fashion and written in standard English?

Reviewer #2: Yes

6. Review Comments to the Author

Reviewer #2: (No Response)

7. PLOS authors have the option to publish the peer review history of their article (what does this mean?). If published, this will include your full peer review and any attached files.

Reviewer #2: Yes: Kevin W. Boyack

---

## [Editor Report · Acceptance letter]

21 Oct 2019

PONE-D-19-16569R1 

Visualizing a field of research: A methodology of systematic scientometric reviews 

Dear Dr. Chen:

I am pleased to inform you that your manuscript has been deemed suitable for publication in PLOS ONE. Congratulations! Your manuscript is now with our production department. 

With kind regards,

on behalf of

Prof. Dr. Wolfgang Glanzel 

Academic Editor

PLOS ONE